# Maximum Variance Unfolding on Disjoint Manifolds

## Abstract

An assumption underlying much of machine learning is that observed data are often sampled from a manifold of much lower dimension than the data space itself. While linear methods such as PCA can often be used to perform dimensionality reduction, they fail to capture nonlinear relationships in the data, which are often present in natural datasets. Maximum variance unfolding is an established and well-studied neighborhood graph-based method for nonlinear dimensionality reduction with the unique property of retaining strong local isometry. However, its applicability on real-world data is limited due to its dependence on the connectivity of the underlying neighborhood graph: in natural datasets, data are often multimodal and lie on disjoint manifolds, giving rise to clusters of points that are distant in the data space. In this work, we present a method that extends MVU to the common case where data lie on disjoint manifolds. We show that it decreases both computation time and memory requirements, and that it improves performance in standard metrics that assess the extent to which the local structure of the data is preserved.

## 1 Introduction

Dimensionality reduction is a vast research field with the fundamental goal of transforming high-dimensional data into a lower-dimensional representation that captures its intrinsic structure. This is related to the manifold assumption, whereby we expect that data are sampled from distributions whose support lies on (or close to) a manifold embedded in the data space (Fefferman et al., 2016).

Despite being widely used as a preprocessing step in machine learning tasks, linear methods for dimensionality reduction, such as principal component analysis (PCA) (Pearson, 1901), are inadequate for capturing nonlinear relationships in the data. While PCA seeks to find a linear subspace that minimizes the reconstruction error of the data points, methods for nonlinear dimensionality reduction generalize this idea to smooth, nonlinear, and lower-dimensional geometries, i.e., manifolds. This is done by constructing a neighborhood graph, where each data point is connected to its $k$-nearest neighbors. Each method then specifies a different problem resulting in embeddings with different properties, e.g., Isomap (Tenenbaum et al., 2000) maps data to a lower dimension while preserving geodesic distances approximated by shortest paths along the neighborhood graph.

Due to their ability to recover nonlinear structure in data, and even in the age of deep learning techniques, nonlinear dimensionality reduction methods are used across many domains and applications including the study of industrial (Wei et al., 2016), chemical (Boninsegna et al., 2015), and biological processes (Dsilva et al., 2018), brain imaging data (Tang et al., 2021), sentiment analysis (Kim & Lee, 2014), remote sensing (Song et al., 2024), facial recognition (Ge et al., 2024), and semi-supervised learning (Pitelis et al., 2014; Huang et al., 2019).

Maximum variance unfolding (MVU) (Weinberger & Saul, 2006a) is a well-studied (Sun et al., 2006; Ghojogh et al., 2021; 2023) method for nonlinear dimensionality reduction which attempts to pull data points apart, effectively "unfolding" them onto the embedding space. MVU has some ineresting properties: unlike kernel-based methods (e.g., kernel PCA (Schölkopf et al., 1998)), it does not require specifying a kernel, and can directly learn the intrinsic structure of the underlying manifold. In some applications, it may be difficult to find an appropriate kernel; e.g., Liu et al. (2014) show that MVU significantly outperforms kernel PCA in industrial process control. Furthermore, MVU is unique among its peers in that it provides local isometry guarantees. In certain

applications, this is a requirement, e.g., Simonetto et al. (2012) use MVU for sensor localization and robotic dispersion problems. Finally, unlike most other methods for nonlinear dimensionality reduction, MVU is immune to the so-called "repeated eigendirection problem", whereby eigenvectors of embedding Jacobians are harmonics of previous ones (Dsilva et al., 2018; Meilă & Zhang, 2024). This is because of MVU's variance maximization objective, which works as a repulsion mechanism which does not allow for the collapse of intrinsic data dimensions.

MVU has been successfully used in many applications and types of data. Weinberger & Saul (2006b) use MVU to recover a $2D$ representation of $60,000$-dimensional text co-occurrence statistics and show that semantic relationships between words are preserved. Mahadevan et al. (2011) develop an extension to MVU which allows it to learn from bimodal data such as EEG-fMRI data and image-text pairs. Wang & Paynabar (2023) use MVU for regression in process optimization. Finally, Song et al. (2007); Wei et al. (2016); Yang & Qi (2024) develop supervised variants of MVU.

However, MVU has two main drawbacks: most importantly, it cannot be applied to data which form a disconnected neighborhood graph. These graphs arise naturally in the common case where data lie on multiple disjoint manifolds (e.g., multimodal data) or simply due to sampling irregularities. Secondly, MVU is computationally expensive, and applying it to datasets with thousands of samples may be prohibitive. In this paper, we propose a simple solution to address the first problem, which involves computing MVU embeddings for disjoint graph components separately, and afterwards reconstructing their global structure. By allowing for parallel computing of MVU on disjoint components, our method also greatly alleviates the second problem, which we demonstrate later. [1]

The rest of the paper is organized as follows: in Section 2, we present MVU and derive its convex relaxation which is used in practice. We also present some of its notable extensions which are relevant for our problem. We present our method in Section 3, and describe and discuss an extensive experimental evaluation in Section 4. We discuss our findings in Section 5 and conclude with some closing remarks in Section 6.

## 2 BACKGROUND

### 2.1 MAXIMUM VARIANCE UNFOLDING

Maximum variance unfolding (MVU) (Weinberger & Saul, 2006a) is a method for nonlinear dimensionality reduction which solves the problem of "unfolding" the data manifold by spreading points in the target space while maintaining local isometry.

Given data $\boldsymbol{X} = \{\boldsymbol{x}_i\}_{i=1}^N, \boldsymbol{x}_i \in \mathbb{R}^D$, we state:

$$\max_{\boldsymbol{y}_1,\ldots,\boldsymbol{y}_N} \quad \sum_{k=1}^N \|\boldsymbol{y}_i\|_2^2 \tag{1}$$

$$\text{s.t.} \quad \|\boldsymbol{y}_i - \boldsymbol{y}_j\|_2^2 = \|\boldsymbol{x}_i - \boldsymbol{x}_j\|_2^2, \quad i \sim j \tag{2}$$

$$\sum_{i=1}^N \boldsymbol{y}_i = \boldsymbol{0} \tag{3}$$

The objective encodes our wish to spread the data as much as possible in the target space, i.e., to maximize the variance of the embeddings $\boldsymbol{y}_1, \ldots, \boldsymbol{y}_N \in \mathbb{R}^d$. With $i \sim j$ indicating a $k$-nearest neighborhood relationship between the $i$-th and $j$-th points, the constraint in equation 2 specifies that distances between neighbors in the target space should be equal to the original distances between those same points. Finally, a centering constraint is enforced in equation 3. This is necessary because if the data were not centered in the target space, the points could be taken indefinitely far away from the origin, maximizing variance but leading to an unbounded problem.

Note that the problem as stated is not convex: the objective consists of maximizing a quadratic function. Furthermore, the bilinear terms on the left hand side of constraint equation 2 define a quadratic equality on the decision variables, which does not, in general, define a convex set. We can,

---

[1]We will provide all the software used to obtain the results presented in this paper as soon as deanonimyzation is allowed.

---

**Algorithm 1** Maximum variance unfolding on disjoint manifolds

---

**Input**: $\boldsymbol{X} \in \mathbb{R}^{(n \times D)}$
**Hyperparameter**: $k \in \mathbb{R}$
**Output**: $\boldsymbol{Y} \in \mathbb{R}^{(N \times d)}$
1: $\mathcal{X}_1, \ldots, \mathcal{X}_C \leftarrow$ build_neighborhood_graph$(\boldsymbol{X}, k)$
2: $\boldsymbol{Y}_1, \ldots, \boldsymbol{Y}_C \leftarrow$ MVU$(\boldsymbol{X}, \mathcal{X}_c), \quad p = 1, \ldots, C$
3: $\mathcal{Z}_1, \ldots, \mathcal{Z}_C \leftarrow$ choose_representative_points$(\boldsymbol{Y}_p), \quad p = 1, \ldots, C$
4: $\mathcal{L} \leftarrow$ choose_intercomponent_connections$(\boldsymbol{X}, \{\mathcal{X}_p\}_{p=1}^C)$
5: $\boldsymbol{Z}_1 \ldots, \boldsymbol{Z}_C \leftarrow$ global_MVU$(\boldsymbol{X}, \{\mathcal{X}_p\}_{p=1}^C, \boldsymbol{Y}, \{\mathcal{Z}_p\}_{p=1}^C, \mathcal{L})$
6: $\boldsymbol{Y}_1, \ldots, \boldsymbol{Y}_C \leftarrow$ translate_components$(\boldsymbol{Y}_p, \mathcal{Z}_p), \quad p = 1, \ldots, C$
7: **Return** $[\boldsymbol{Y}_1 \quad \cdots \quad \boldsymbol{Y}_C]$

---

however, reach a convex version of this problem through the substitution $\boldsymbol{K} = \boldsymbol{Y}^\top \boldsymbol{Y} \in \mathbb{R}^{N \times N}$ (intermediate steps and further details can be found in the Appendix):

$$\max_{\boldsymbol{K}} \quad \mathrm{tr}(\boldsymbol{K}) \tag{4}$$

$$\text{s.t.} \quad \boldsymbol{K}_{ii} - 2\boldsymbol{K}_{ij} + \boldsymbol{K}_{jj} = \|\boldsymbol{x}_i - \boldsymbol{x}_j\|_2^2, \quad i \sim j \tag{5}$$

$$\boldsymbol{K} \succeq 0, \tag{6}$$

$$\sum_{i=1}^N \sum_{j=1}^N \boldsymbol{K}_{ij} = 0. \tag{7}$$

The SDP in MVU is solved using interior point methods, whose per-iteration computational complexity is $O((kN)^3)$ and the memory requirement is $O((kN)^2)$ (Borchers & Young, 2007). This cost is prohibitive for datasets with more than a few thousand samples, where convergence may take a long time or the data may not fit in memory at all, making MVU difficult to apply to real world data.

## 2.2 EXTENSIONS TO MVU

An issue with all neighborhood graph-based methods for nonlinear dimensionality reduction is that they fail when neighborhood graphs are not connected, i.e., when disconnected components arise from the $k$-nearest neighbor selection. In the case of MVU, if there are disjoint components, the problem becomes unbounded as components could be taken arbitrarily far away from the origin, increasing variance to infinity.

A simple solution to this is the one employed in Van Der Maaten et al. (2009): starting with the largest component, we find the component that's closest to it and create a connection between the closest points in each component. Those are now considered only one component, and this process is repeated until the neighborhood graph is connected. Another solution proposed for dealing with this limitation, called the enhanced neighborhood graph (ENG), was proposed by Fan et al. (2018).

Finally, some extensions may be used to alleviate the computational burden of MVU. The Nyström approximation (Platt, 2005) is a low-rank matrix approximation used on kernel methods. It can be applied to MVU on disjoint components by embedding the largest component as usual, and projecting the rest of the data points using a Gaussian kernel. Landmark MVU (Weinberger et al., 2005) embeds a set of randomly sampled points, so-called "landmarks", and computes the embeddings for the remaining points as linear combinations of the landmarks. However, by ignoring local structure, both of these methods sacrifice the strong local isometry guarantees of MVU.

## 3 MAXIMUM VARIANCE UNFOLDING ON DISJOINT MANIFOLDS

In this section, we describe our method in detail, which we summarize in pseudocode in Algorithm 1. We provide a notation (glossary? variable index?) table in Appendix TODO. The steps of our algorithm, given data $\boldsymbol{X} \in \mathbb{R}^{N \times D}$ and hyperparameter $k \in \mathbb{R}$, are as follows:

1. A neighborhood graph is built based on the $k$-nearest neighbors of each point. The components are then found by simply starting a breadth-first search on unvisited nodes until all nodes have been visited. The indices of the points that belong to each component are collected into sets $\mathcal{X}_1, \ldots, \mathcal{X}_C$, where $C$ is the number of components. Note that:

   - The above definition implies $\mathcal{X}_p \cap \mathcal{X}_q = \varnothing, p \neq q, p, q = 1, \ldots, C$ (components are disjoint) and $\cup_{p=1}^{C} \mathcal{X}_p = [N]$ (the union of components is a collection of the indices of all the data points).
   - The number of components varies for the choice of $k$. The smaller the value of this hyperparameter, the larger the amount of components, and vice versa. We can always set the number of components to 1 by choosing a sufficiently large $k$.

2. Maximum variance unfolding (MVU) is applied to each component separately, yielding embedded data $\boldsymbol{Y}_p \in \mathbb{R}^{|\mathcal{X}_p| \times d_p}$. Importantly, these computations can be done in parallel, since no intercomponent connections are considered at this time. We note that the dimensionality of the computed embeddings $d_p$ depends on the intrinsic dimensionality of the corresponding manifold: like MVU, we retain the top eigenvalues of $\boldsymbol{K}$ which preserve some percentage of the variance in the original data. Furthermore, each component $\boldsymbol{Y}_p$ is zero-centered, and not yet in its final position.

3. For each embedded component $\boldsymbol{Y}_p$, a set of "representative points" $\mathcal{Z}_p$ is chosen. The goal of this step is to choose a subset of the points of each embedded component that is a good estimate of its global geometry. We experiment with two methods for this selection, which we detail in Section 3.1. In either case, the number of representative points selected for each component is twice the dimensionality of that component, i.e., $|\mathcal{Z}_p| = 2d_p$. We can generally expect the number of representative points to be much smaller than the size of the component, i.e., $2d_p << |\mathcal{X}_p|$.

4. Connections are created between components until the neighborhood graph is connected. This is done in the same way as described in Section 2.1 for vanilla MVU: we find the largest component and the component closest to it, create a connection between the closest points between those two components, and treat them as a single component. This is done iteratively until the neighborhood graph is connected. Creating these connections amounts to saving the indices of the points that share a connection in pairs, for use in the next step. The points of each component that are selected as part of intercomponent connections are also added to the set of representative points of the respective components. This step is done in the sample space and could be performed earlier in the algorithm, but we present it here since it is related to the next step.

5. We perform a final "global MVU" on the representative points of each component: we preserve isometry between all representative points of a component (intracomponent connections) and the connections created in the previous step (intercomponent connections). This step takes the representative points of the zero-centered components $\boldsymbol{Y}_p$, given by the corresponding index subsets $\mathcal{Z}_p$, and puts them in their final location, yielding $\boldsymbol{Z}_p \in \mathbb{R}^{|\mathcal{Z}_p| \times d}$. To ensure that all components have the same embedding dimensionality, we add dimensions (zeroes) when required such that $d = \max d_p, \ p = 1, \ldots, C$ before this step.

6. The remaining points of each component $\boldsymbol{Y}_p$ are translated to their positions relative to the representative points of that component $\boldsymbol{Z}_p$. This is done by representing the points in homogeneous coordinates and computing an affine transformation matrix.

   It is guaranteed that, in the linearized global MVU, there are the same or more dimensions to define each component, assuring no loss of information. Then, we can compute an affine transformation that takes the representative points from each component into the global MVU. Subsequently, the results obtained from applying each transformation to its respective component are aggregated.

Further details about the computations used in our method can be found in the Appendix. We explain the need and method for selecting representative points of a component in Section 3.1, and the "global MVU" step in greater detail in Section 3.2.

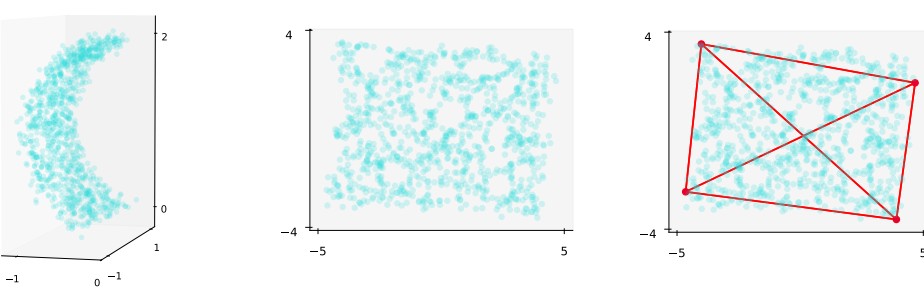

(a) Component in the data space $\mathbb{R}^3$    (b) Component unfolded in $\mathbb{R}^2$    (c) Representative points

Figure 1: Illustration of the process of selecting representative points of a component. (a) Component in ambient (data) space, (b) Embedded component and principal directions, (c) Selected representative points and their connections highlighted.

## 3.1 Choosing Sets of Representative Points

After embedding the components separately, the goal is to translate each of them to their final position in the target space; this is done in the final three steps of the algorithm. However, performing the "global MVU" step on all the points would partly defeat the purpose of embedding each component separately, as optimizing the entire Gramian would incur the computational and memory requirements described in Section 2.1.

Instead, we propose working with only a small subset of the points in each component. We would like the chosen subset to be a good approximation of the structure formed by the points of each component. Since embedded components are unfolded to their maximum variance, we can expect the convex hull defined by the extrema along their principal directions to be a good approximation of that structure. This is illustrated in Figure 3.

Thus, our procedure to select the representative subset of each component is to represent its points in terms of its principal components (computed through SVD, which is computationally cheap), and select the indices of the points which are the maxima and minima along each principal direction. Given an embedded component of dimension $d_p \in \mathbb{N}$, we have $|\mathcal{Z}_p| = 2d_p$. However, recall that points selected as part of intercomponent connections (step 4) are also added to the set of representative points of a component.

## 3.2 Global MVU

Having selecting the representative subset of each component, we choose which intercomponent connections to keep as described in step 4. We also need to make sure that all embedded components have the same dimensionality; to this end, we add zero-filled dimensions as required such that $d_p \leftarrow \max\{d_q, q = 1, \ldots, C\}, p = 1, \ldots, C$, that is, all embedded components have the same dimensionality as the embedded component with highest dimensionality.

We can now formulate our "global MVU" step, which is illustrated in Figure 2:

$$\max_{\boldsymbol{z}_{1,1},\ldots,\boldsymbol{z}_{p,|\mathcal{Z}_p|} \in \mathbb{R}^{d_p}} \quad \sum_{p=1}^{C} \sum_{i=1}^{|\mathcal{Z}_p|} \|\boldsymbol{z}_{p,i}\|_2^2 \tag{8}$$

$$\text{s.t.} \quad \|\boldsymbol{z}_{p,i} - \boldsymbol{z}_{p,j}\|_2^2 = \|\boldsymbol{y}_{p,i} - \boldsymbol{y}_{p,j}\|_2^2, \quad p = 1, \ldots, C; \ i, j \in \mathcal{Z}_p \tag{9}$$

$$\|\boldsymbol{z}_{p,i} - \boldsymbol{z}_{q,j}\|_2^2 = \|\boldsymbol{x}_{p,i} - \boldsymbol{x}_{q,j}\|_2^2, \quad (\boldsymbol{x}_{p,i}, \boldsymbol{x}_{q,j}) \text{ i.c. connections} \tag{10}$$

$$\sum_{p=1}^{C} \sum_{i=1}^{|\mathcal{Z}_p|} \boldsymbol{z}_{p,i} = \boldsymbol{0} \tag{11}$$

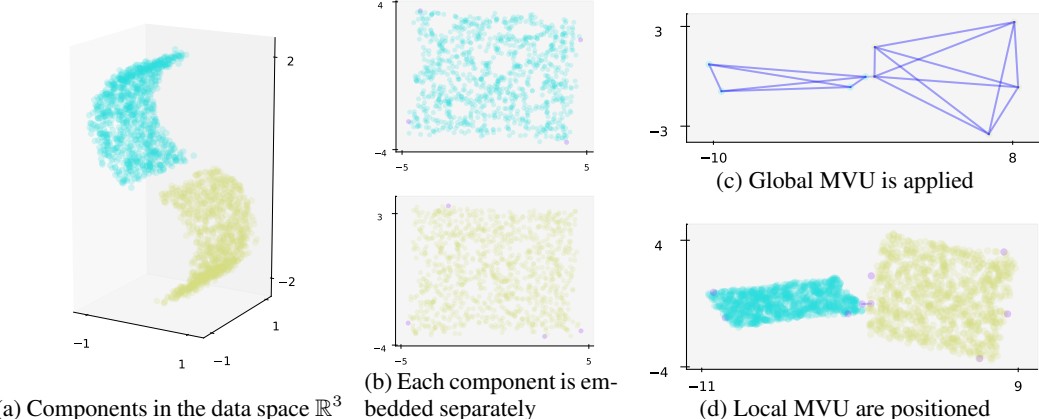

(a) Components in the data space $\mathbb{R}^3$

(b) Each component is embedded separately

(c) Global MVU is applied

(d) Local MVU are positioned

Figure 2: Illustration of the "global MVU" step. (a) Two components in ambient (data) space, (b) Each component is embedded separately, and their representative subsets are computed, (c) Global MVU is applied, retaining both intercomponent connections and distances between all representative points within the same component, (d) The remaining points of each component are translated to their final positions.

The objective (Equation 8) is the same as in vanilla MVU: to maximize the variance. However, in this case, we are only spreading out the set of representative points of all components. Equation 9 defines intracomponent isometry constraints: all distances between points in each set of representative points should be kept. Finally, Equation 10 defines intercomponent isometry constraints based on the connections selected in step 4. Note that in these constraints, isometry is with respect to the distances in the original data space. As usual, we zero-center the solution to avoid unboundedness. We note that the problem as stated is not convex, but turning it into a convex problem is done in the same manner as for MVU (cf. Section 2.1).

## 4 EVALUATION

In this section, we describe in detail the experimental evaluation performed to validate our method. We start by enumerating the methods we test in our benchmarks in Section 4.1. Then, we present a diverse set of both artificial and natural datasets in Section 4.2, which we use to benchmark our methods. We evaluate the performance of all methods on all datasets according to the metrics described in Section 4.3.

### 4.1 METHODS

In addition to comparing our method, which we call maximum variance unfolding on disjoint manifolds (MVU-DM), with vanilla MVU (Weinberger & Saul, 2006a), we consider the following representative methods for nonlinear dimensionality reduction: kernel PCA (KPCA) (Schölkopf et al., 1998), Laplacian eigenmaps (LE) (Belkin & Niyogi, 2003), locally linear embedding (LLE) (Roweis & Saul, 2000), Hessian locally linear embedding (HLLE) (Donoho & Grimes, 2003), Isomap (Tenenbaum et al., 2000), and local tangent space alignment (Zhang & Zha, 2004) (LTSA). We also add the enhanced neighborhood graph (ENG) (Fan et al., 2018) to Isomap, which performed best in their experiments.

Despite being the most commonly used method for data visualization, t-SNE (Maaten & Hinton, 2008) is expected to perform poorly in our benchmarks as it emphasizes clustering over preserving distances between points. As such, we exclude it from our experiments.

## 4.2 DATASETS

We include a variety of both artificial and natural datasets with varying global structures, scales, and intrinsic dimensionalities. We describe them here on a high level, and leave a presentation of all the details to the Appendix.

### 4.2.1 ARTIFICIAL DATASETS

We considered all datasets from Van Der Maaten et al. (2009) and Fan et al. (2018). However, we excluded the fully connected datasets for not fitting the objective of this study. Additionally, we exclude the Broken Swiss Roll dataset since it is largely redundant when compared with the Broken S-curve dataset in terms of their properties.

The selection of artificial datasets we use focuses on evaluating the way each method relates components that were found disconnected from the neighborhood graph. So, we consider the Broken S-curve (BSC) dataset, consisting of 4 sections of a bent 2D manifold forming a 3D 'S' structure. We utilized two variations of the Swiss Roll: one consists of two distinct Swiss rolls separated by an arbitrary distance (SR1), while the other features two non-colliding Swiss rolls placed adjacent to each other (SR2). The last synthetic dataset considered is the Four Moons (FM) dataset which consists of two pairs of C-shaped manifolds. Each pair is composed of a smaller manifold nested within a larger one, with the two pairs positioned parallel to each other.

All of these artificial datasets were generated with 2000 points, to which we added Gaussian noise with variance 0.05. Formulas for their generation and illustrations may be found in the appendix.

### 4.2.2 NATURAL DATASETS

We used all the disconnected natural datasets from Van Der Maaten et al. (2009) and Fan et al. (2018). The COIL20 dataset consists of 1440 single-channel ($128 \times 128$) photos of 20 different objects, taken from varying angles of rotation. ORL is a 400-photo dataset of faces from 40 distinct subjects, from different angles, with ($112 \times 92$) resolution. The MIT-CBCL dataset also consists of photos of subjects' faces taken from different angles. It comprises 2059 ($64 \times 64$) photos. Additionally, the Olivetti dataset comprises the same original data as the ORL dataset; however, it was treated and is made available by scikit-learn. Although they have the same images, they are 64 by 64 pixels.

## 4.3 METRICS

We follow Van Der Maaten et al. (2009) and assess all methods according to the quality of their embeddings with respect to 1-nearest neighbor classification performance (cf. Sanguinetti (2008)), trustworthiness, and continuity (Venna & Kaski, 2006). These metrics evaluate to what extent the local structure of the data is preserved in the embeddings.

The 1-nearest neighbor classifier error is the percentage of points whose closest neighbor in the embedding space is of a different class than in the original space (Sanguinetti, 2008). Classes are assigned to points according to hypercubes defined in the data space.

The trustworthiness and continuity assess how well neighborhoods around each point are preserved in the embedding space (Venna & Kaski, 2006). For the $i$-th point, $\mathcal{U}_i^k$ is the set of its $k$-nearest neighbors in the embedding space. With $j$ its $r(i, j)$-nearest neighbor in the input space, trustworthiness is given by:

$$T(k) = 1 - \frac{2}{Nk(2N - 3k - 1)} \sum_{i=1}^{N} \sum_{j \in \mathcal{U}_i^k} \max(0, r(i, j) - k) \qquad (12)$$

We can think of $r(i, j)$ as a ranking: a list, ordered by distance, of the neighbors of $i$ in the original space. Then, since $\mathcal{U}_i^k$ is the set of neighbors of $i$ in the embedding space, trustworthiness penalizes "intrusions" into the set of nearest neighbors after embedding.

Analogously, if we define $\mathcal{V}_i^k$ as the set of the $k$-nearest neighbors of $i$ in the original space, and $\hat{r}(i, j)$ as the ranking of $j$ in terms of nearest-neighbors of $i$ in the embedding space, we get the

Table 1: 1-NN results (Smaller values are better)

| | Artificial Datasets | | | | Natural Datasets | | | |
|---|---|---|---|---|---|---|---|---|
| | BSC | SR1 | SR2 | FM | COIL20 | ORL | MIT-CBCL | Olivetti |
| Isomap | 6.50% | 15.30% | 9.15% | **0.00%** | 5.83% | 11.25% | 1.60% | 18.25% |
| Isomap+ENG | 13.10% | 15.15% | 8.70% | 4.95% | 7.36% | 11.75% | 1.65% | 18.25% |
| LLE | **4.15%** | 26.75% | 26.45% | **0.00%** | 7.43% | 9.00% | 1.70% | 14.75% |
| HLLE | 5.20% | **7.55%** | 8.05% | 0.10% | 7.29% | 25.75% | 2.43% | 20.50% |
| LE | 5.05% | 32.15% | 32.25% | 1.05% | 10.35% | 13.25% | 1.99% | 31.00% |
| LTSA | 9.20% | 11.90% | **7.55%** | 0.15% | 7.01% | 25.75% | 2.38% | 37.25% |
| K-PCA | 50.45% | 26.75% | 16.85% | **0.00%** | 5.83% | **4.00%** | **1.41%** | 13.25% |
| MVU | 6.70% | 13.35% | 13.10% | **0.00%** | 5.69% | 10.75% | 1.89% | 14.50% |
| MVU-DM | 4.85% | 9.20% | 9.75% | 2.65% | **4.38%** | 6.25% | 1.94% | **8.25%** |

Table 2: Trustworthiness results (Larger values are better)

| | Artificial Datasets | | | | Natural Datasets | | | |
|---|---|---|---|---|---|---|---|---|
| | BSC | SR1 | SR2 | FM | COIL20 | ORL | MIT-CBCL | Olivetti |
| Isomap | 99.18% | 98.05% | 99.76% | 99.10% | 99.09% | 98.69% | 99.67% | 97.16% |
| Isomap+ENG | 98.01% | 98.12% | 99.93% | 97.27% | 98.26% | 98.46% | 99.42% | 97.16% |
| LLE | **99.53%** | 94.81% | 94.92% | 99.17% | 97.99% | 95.86% | 99.06% | 91.16% |
| HLLE | 98.85% | 99.81% | 99.95% | 98.68% | 97.91% | 90.73% | 99.06% | 88.92% |
| LE | 98.94% | 93.76% | 93.69% | 99.72% | 98.56% | 98.20% | 99.73% | 94.03% |
| LTSA | 97.58% | 99.41% | **99.96%** | 98.39% | 96.98% | 90.73% | 99.20% | 88.52% |
| K-PCA | 92.90% | 92.19% | 89.41% | **100.00%** | **99.43%** | **99.37%** | **99.90%** | **98.45%** |
| MVU | 97.99% | 98.44% | 97.32% | 98.64% | 97.86% | 97.54% | 99.33% | 97.03% |
| MVU-DM | 99.50% | **99.90%** | 99.70% | 97.62% | 99.10% | 98.10% | 99.10% | 98.30% |

metric of continuity, which measures how many neighbors of $i$ in the data space are no longer its neighbors in the embedding space:

$$C(k) = 1 - \frac{2}{Nk(2N - 3k - 1)} \sum_{i=1}^{N} \sum_{j \in \mathcal{V}_i^k} \max(0, \hat{r}(i, j) - k) \qquad (13)$$

## 5 DISCUSSION

Our results corroborate those on the seminal analysis by Van Der Maaten et al. (2009): MVU is often among the best performing methods, particularly on the artificial datasets. In those scenarios, our method tended to improve performance over the baseline MVU. However, improvements were even more robust in the natural datasets, where our proposed MVU-DM achieved some of the best results.

Besides generally improving performance over MVU, we find that our method significantly speeds up execution. We present speedups of MVU-DM over vanilla MVU in Table 4 for different values of $k$.

## 6 CONCLUSION

We introduced a new method for applying MVU to data that lie on disjoint manifolds, or which display a disconnected neighborhood graph for any reason. Our experiments on a variety of both artificial and natural datasets show that its ability to preserve local structure is at least as good as that of MVU, while both improving its efficiency and increasing its applicability to disconnected neighborhood graphs. Furthermore, our method does not require any additional hyperparameters to vanilla MVU, making it more adequate for cross-validation.

Table 3: Continuity results (Larger values are better)

| | Artificial Datasets | | | | Natural Datasets | | | |
|---|---|---|---|---|---|---|---|---|
| | BSC | SR1 | SR2 | FM | COIL20 | ORL | MIT-CBCL | Olivetti |
| Isomap | **99.85%** | 99.55% | 99.91% | 99.55% | **99.80%** | 99.68% | 99.87% | 99.41% |
| Isomap+ENG | 99.60% | 99.55% | **99.95%** | 99.60% | 99.64% | 99.66% | 99.77% | 99.37% |
| LLE | 98.49% | 99.40% | 99.40% | 98.68% | 99.14% | 97.30% | 99.31% | 92.47% |
| HLLE | 95.70% | 95.69% | 95.80% | 93.50% | 98.76% | 94.86% | 98.60% | 88.65% |
| LE | 96.86% | 99.40% | 99.35% | 80.30% | 99.06% | 99.16% | 99.63% | 96.80% |
| LTSA | 87.55% | 93.10% | 95.59% | 93.82% | 99.11% | 94.86% | 98.52% | 88.65% |
| K-PCA | 99.28% | 98.78% | 98.03% | **100.00%** | **99.80%** | 99.48% | **99.91%** | 99.12% |
| MVU | 99.24% | 99.58% | 99.76% | 99.45% | 99.73% | **99.71%** | 99.82% | 99.59% |
| MVU-DM | 99.80% | **99.90%** | 99.80% | 99.21% | **99.80%** | 99.70% | 99.80% | **99.70%** |

Table 4: Time speedup of MVU-DM compared to MVU for different values of $k$

| | Artificial Datasets | | | | Natural Datasets | | | |
|---|---|---|---|---|---|---|---|---|
| | BSC | SR1 | SR2 | FM | COIL20 | ORL | MIT-CBCL | Olivetti |
| $k = 5$ | 6.69 | 3.24 | 2.43 | 6.10 | 4.02 | 1.15 | 7.96 | 0.55 |
| $k = 10$ | 12.18 | 6.40 | 6.61 | 12.81 | 3.03 | 1.09 | 4.54 | 1.23 |
| $k = 15$ | 11.13 | 4.14 | 5.67 | 15.94 | 1.59 | 1.09 | 2.48 | 1.39 |

ACKNOWLEDGMENTS

This work was partly funded by the Foundation of Science and Technology through scholarship 2024.04726.BD.

REPRODUCIBILITY STATEMENT

We have prepared a codebase with all the software necessary to reproduce the experiments in this paper. This includes all the nonlinear dimensionality methods used in the experiments, all extensions, metrics, and datasets.

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

## A   METHODS FOR NONLINEAR DIMENSIONALITY REDUCTION

Practically all methods for nonlinear dimensionality reduction use neighborhood graphs, which are built by connecting each point to its $k$-nearest neighbors according to some metric (usually the Euclidean distance). The idea is that while it is difficult to define some global objective for how points should be arranged in the embedding space, we can use information about the local structure around each data point. Many methods for nonlinear dimensionality reduction involve finding, for some data $\{\boldsymbol{x}_i\}_{i=1}^N \in \mathbb{R}^D$, embeddings $\{\boldsymbol{y}_i\}_{i=1}^N \in \mathbb{R}^d$ with $d << D$ such that:

$$\min_{\boldsymbol{y}_1,\ldots,\boldsymbol{y}_N} \sum_{i=1}^N \sum_{j=1}^N (d(\boldsymbol{x}_i, \boldsymbol{x}_j) - \|\boldsymbol{y}_i - \boldsymbol{y}_j\|_2)^2 \tag{14}$$

If we set $d(\cdot, \cdot)$ to be the Euclidean distance, we recover classical multidimensional scaling, a linear method for dimensionality reduction that is equivalent to PCA Torgerson (1952).

### A.1   ISOMAP

Isomap (Tenenbaum et al., 2000) is a particular formulation of classical multidimensional scaling (Torgerson, 1952), which we repeat here:

$$\min_{\boldsymbol{y}_1,\ldots,\boldsymbol{y}_N} \sum_{i=1}^N \sum_{j=1}^N (d(\boldsymbol{x}_i, \boldsymbol{x}_j) - \|\boldsymbol{y}_i - \boldsymbol{y}_j\|_2)^2 \tag{15}$$

Given a dataset $\{\boldsymbol{x}_i\}_{i=1}^N \in \mathbb{R}^D$, the above expresses our wish to find embeddings $\{\boldsymbol{y}_i\}_{i=1}^N \in \mathbb{R}^d$ such that some distance $d(\cdot, \cdot)$ is maintained between all points when embedded. In particular, Isomap constructs a $k$-neighborhood graph, from which it computes shortest-path distances $\Delta_{ij}$ between each pair of points. These distances correspond to approximate geodesics along the data manifold, and can be computed with Dijsktra's (Dijkstra, 2022) or Floyd-Warshall's (Floyd, 1962) algorithms.

Isomap (Tenenbaum et al., 2000) builds a distance matrix of approximated geodesics between all points, where geodesics are estimated as shortest-distance paths across the neighborhood graph. Then, it minimizes the above objective where $d(\boldsymbol{x}_i, \boldsymbol{x}_j)$ is the approximated geodesic distance $\Delta_{ij}$ between the $i$-th and $j$-th points.

However, we may also perform Isomap in closed form. We can retrieve the inner product matrix, i.e., the Gramian of the embedded data from $\Delta_{ij}$ via "double-centering":

$$\boldsymbol{K} = -\frac{1}{2}(\mathbb{I} - \frac{1}{n}\boldsymbol{e}\boldsymbol{e}^\top)\boldsymbol{\Delta}^2(\mathbb{I} - \frac{1}{n}\boldsymbol{e}\boldsymbol{e}^\top), \tag{16}$$

with $\boldsymbol{e} = [1, \ldots, 1]^\top \in \mathbb{R}^n$. Then, from the eigendecomposition of $\boldsymbol{K} = \boldsymbol{Q}\boldsymbol{\Lambda}\boldsymbol{Q}^\top$, we recover the embeddings:

$$\boldsymbol{Y} = \sqrt{\boldsymbol{\Lambda}_d}\boldsymbol{Q}_d^\top, \tag{17}$$

where $\boldsymbol{\Lambda}_d$ and $\boldsymbol{Q}_d^\top$ contain the $d$ largest eigenvalues and eigenvectors respectively.

### A.2   LOCALLY LINEAR EMBEDDING

While Isomap tries to preserve geodesic distances globally across the manifold, locally linear embedding (LLE) (Roweis & Saul, 2000) attempts to preserve only local properties of the data. By assuming that the $k$-neighborhood $\mathcal{N}_i^k$ around each point $\boldsymbol{x}_i$ lies on a linear patch of the manifold, LLE starts by defining each (high-dimensional) point as a linear combination of its $k$-nearest neighbors and finding the corresponding reconstruction weights $W \in \mathbb{R}^{D \times D}$:

$$\min_W \quad \sum_i^N \left\| \boldsymbol{x}_i - \sum_j w_{ij}\boldsymbol{x}_j \right\|_2^2, i \sim j \tag{18}$$

$$\text{s.t.} \quad \sum_{j=1}^N w_{ij} = 1, \quad i = 1, \ldots, N \tag{19}$$

where $i \sim j$ indicates a nearest-neighbor relation.

Then, the goal is to find projections $\{\boldsymbol{y}_i\}_{i=1}^N \in \mathbb{R}^d$ such that each projected point can be reliably computed as a linear combination of its $k$-nearest neighbors in the original space using the reconstruction weights above. We can then formulate the LLE objective:

$$\min_{\boldsymbol{y}_i} \quad \sum_i \left\| \boldsymbol{y}_i - \sum_j w_{i,j} \boldsymbol{y}_{i,j} \right\|_2^2 \tag{20}$$

### A.3 HESSIAN LLE

Hessian LLE (HLLE) (Donoho & Grimes, 2003) extends LLE by computing a global Hessian matrix describing the manifold's curvature, and minimizing it. This matrix is computed from the factorization of each local patch around a data point into principal directions. After merging these and a column of ones, this output is orthogonalised in a Gram-Schmidt manner to estimate the Local Hessian.

Finishing by applying the eigenvalue decomposition on the estimated Hessian matrix $\mathcal{H}$:

$$\mathcal{H}\boldsymbol{\alpha} = \lambda\boldsymbol{\alpha} \tag{21}$$

and selecting the $d$ smallest eigenvalues and their associated eigenvectors, to define the reduced space.

### A.4 LAPLACIAN EIGENMAPS

Laplacian Eigenmaps (LE) (Belkin & Niyogi, 2003) compute edge weights $w_{ij}$ between $k$-nearest neighbors using the Gaussian kernel function:

$$w_{i,j} = e^{-\frac{\|\boldsymbol{x}_i - \boldsymbol{x}_j\|_2^2}{2\sigma^2}}, \quad i \sim j \tag{22}$$

Since larger weights $w_{ij}$ correspond to smaller distances in the original space, we try to put points that are nearby in the data space as close as possible in the embedding space:

$$\min_{\boldsymbol{y}_i} \quad \sum_{i,j} w_{i,j} \|\boldsymbol{y}_i - \boldsymbol{y}_j\|_2^2, \quad i \sim j \tag{23}$$

### A.5 LOCAL TANGENT SPACE ALIGNMENT

Local Tangent Space Alignment (LTSA) (Zhang & Zha, 2004) aligns local tangent spaces to preserve the local geometric structure. For each point, it computes a local tangent space using PCA on its neighborhood, then finds a global embedding that best aligns these local tangent spaces. The global tangent space matrix is $\boldsymbol{B}$ built iteratively:

$$\boldsymbol{B}_{\mathcal{N}_i \mathcal{N}_i} = \boldsymbol{B}_{\mathcal{N}_{i-1} \mathcal{N}_{i-1}} + \boldsymbol{J}_k (\boldsymbol{I} - \boldsymbol{V}_i \boldsymbol{V}_i^\top) \boldsymbol{J}_k, \tag{24}$$

where $\mathcal{N}_i$ define the neighborhood indexes of $i$, and $\boldsymbol{J}_k(\boldsymbol{I} - \boldsymbol{V}_i \boldsymbol{V}_i^\top)\boldsymbol{J}_k$ is the double centered PCA projections $\boldsymbol{V}_i \boldsymbol{V}_i^\top$.

### A.6 KERNEL PCA

Finally, we mention kernel PCA (KPCA) (Schölkopf et al., 1998), which does not rely on neighborhood graphs. Instead, the user picks a kernel function $k(\boldsymbol{x}_i, \boldsymbol{x}_j)$ with which the data points are mapped to a higher-dimensional feature space, where PCA is applied. Embeddings are found by solving an eigenvalue problem:

$$\boldsymbol{K}\boldsymbol{\alpha} = \lambda\boldsymbol{\alpha} \tag{25}$$

where $\boldsymbol{K}$ is the kernel matrix with entries $k_{ij} = k(\boldsymbol{x}_i, \boldsymbol{x}_j)$.

# B CONVEX MVU

In this Section we derive a convex version of MVU from the original formulation, which we reiterate:

$$\max_{\boldsymbol{y}_1,\ldots,\boldsymbol{y}_N} \quad \sum_{i=1}^{N} \|\boldsymbol{y}_i\|_2^2 \tag{26}$$

$$\text{s.t.} \quad \|\boldsymbol{y}_i - \boldsymbol{y}_j\|_2^2 = \|\boldsymbol{x}_i - \boldsymbol{x}_j\|_2^2, \quad i \sim j \tag{27}$$

$$\sum_{i=1}^{N} \boldsymbol{y}_i = \boldsymbol{0} \tag{28}$$

Again, we not that this is not a convex problem since the objective is a maximization of a convex function and the constrain encoded in 27 does not, in general, define a convex set. Expanding the squared terms:

$$\max_{\boldsymbol{y}_1,\ldots,\boldsymbol{y}_n} \quad \sum_{i=1}^{n} \boldsymbol{y}_i^\top \boldsymbol{y}_i \tag{29}$$

$$\text{s.t.} \quad \boldsymbol{y}_i^\top \boldsymbol{y}_i - 2\boldsymbol{y}_i^\top \boldsymbol{y}_j + \boldsymbol{y}_j^\top \boldsymbol{y}_j = \|\boldsymbol{x}_i - \boldsymbol{x}_j\|_2^2, \quad i \sim j \tag{30}$$

$$\sum_{i,j=1}^{n} \boldsymbol{y}_i^\top \boldsymbol{y}_j = 0 \tag{31}$$

By collecting the embedded points into the columns of a matrix $\boldsymbol{Y} = [\boldsymbol{y}_1 \quad \cdots \quad \boldsymbol{y}_n] \in \mathbb{R}^{d \times N}$, we can rewrite the problem:

$$\max_{\boldsymbol{Y}} \quad \text{tr}(\boldsymbol{Y}^\top \boldsymbol{Y}) \tag{32}$$

$$\text{s.t.} \quad \boldsymbol{e}_i^\top \boldsymbol{Y}^\top \boldsymbol{Y} \boldsymbol{e}_i - 2\boldsymbol{e}_i^\top \boldsymbol{Y}^\top \boldsymbol{Y} \boldsymbol{e}_j + \boldsymbol{e}_j^\top \boldsymbol{Y}^\top \boldsymbol{Y} \boldsymbol{e}_j = \|\boldsymbol{x}_i - \boldsymbol{x}_j\|_2^2, \quad i \sim j \tag{33}$$

$$\boldsymbol{1}^\top \boldsymbol{Y}^\top \boldsymbol{Y} \boldsymbol{1} = 0, \tag{34}$$

where $\boldsymbol{e}_i$ is a "selection" or "one-hot" vector, i.e., its elements are all zero except for the element at the $i$-th index, which is 1. Note that, now, the whole problem depends on $\boldsymbol{Y}^\top \boldsymbol{Y}$, so we introduce a new variable $\boldsymbol{K} = \boldsymbol{Y}^\top \boldsymbol{Y} \in \mathbb{R}^{N \times N}$ to linearize the terms that depend on $\boldsymbol{Y}^\top \boldsymbol{Y}$:

$$\max_{\boldsymbol{K},\boldsymbol{Y}} \quad \text{tr}(\boldsymbol{K}) \tag{35}$$

$$\text{s.t.} \quad \boldsymbol{e}_i^\top \boldsymbol{K} \boldsymbol{e}_i - 2\boldsymbol{e}_i^\top \boldsymbol{K} \boldsymbol{e}_j + \boldsymbol{e}_j^\top \boldsymbol{K} \boldsymbol{e}_j = \|\boldsymbol{x}_i - \boldsymbol{x}_j\|_2^2, \quad i \sim j \tag{36}$$

$$\boldsymbol{1}^\top \boldsymbol{K} \boldsymbol{1} = 0 \tag{37}$$

$$\boldsymbol{K} = \boldsymbol{Y}^\top \boldsymbol{Y} \tag{38}$$

Because $\boldsymbol{K} = \boldsymbol{Y}^\top \boldsymbol{Y}$ defines an inner product matrix, we can replace that constraint if we make sure that $\boldsymbol{K}$ is both symmetric, positive semidefinite and that the rank of $\boldsymbol{K}$ is not greater than the dimension of the $\boldsymbol{y}_i$s:

$$\max_{\boldsymbol{K}} \quad \text{tr}(\boldsymbol{K}) \tag{39}$$

$$\text{s.t.} \quad \boldsymbol{e}_i^\top \boldsymbol{K} \boldsymbol{e}_i - 2\boldsymbol{e}_i^\top \boldsymbol{K} \boldsymbol{e}_j + \boldsymbol{e}_j^\top \boldsymbol{K} \boldsymbol{e}_j = \|\boldsymbol{x}_i - \boldsymbol{x}_l\|_2^2, \quad i \sim j \tag{40}$$

$$\boldsymbol{1}^\top \boldsymbol{K} \boldsymbol{1} = 0 \tag{41}$$

$$\boldsymbol{K} \succeq 0 \tag{42}$$

$$\text{rk}(\boldsymbol{K}) \leq n \tag{43}$$

Now, the only nonconvexity in the problem is given by the rank constraint. If we remove it, we arrive at a convex relaxation of the original problem, which is the final formulation of MVU:

$$\max_{\boldsymbol{K}} \quad \text{tr}(\boldsymbol{K}) \tag{44}$$

$$\text{s.t.} \quad \boldsymbol{e}_i^\top \boldsymbol{K} \boldsymbol{e}_i - 2\boldsymbol{e}_i^\top \boldsymbol{K} \boldsymbol{e}_j + \boldsymbol{e}_j^\top \boldsymbol{K} \boldsymbol{e}_j = \|\boldsymbol{x}_i - \boldsymbol{x}_j\|_2^2, \quad i \sim j \tag{45}$$

$$\mathbf{1}^\top \boldsymbol{G} \mathbf{1} = 0 \tag{46}$$

$$\boldsymbol{K} \succeq 0 \tag{47}$$

Here, $\boldsymbol{K} \in \mathbb{R}^{N \times N}$ is a Gramian (or inner product) matrix, so that maximizing its trace corresponds to maximizing the variance of the data in the target space.

## C    DATASETS

### C.1    ARTIFICIAL DATASETS

In the following datasets we use a small Gaussian perturbation $\epsilon \in \mathcal{N}(0, 0.05)$ added to the generated data points:

**Broken S-curve**

The dataset is defined by:

$$t_0 \sim \mathcal{U}\left(-\frac{3\pi}{2}, -\frac{3\pi}{2} + 0.4\pi\right) \tag{48}$$

$$t_1 \sim \mathcal{U}\left(-\frac{3\pi}{2} + 0.6\pi, -\frac{3\pi}{2} + 1.4\pi\right) \tag{49}$$

$$t_2 \sim \mathcal{U}\left(-\frac{3\pi}{2} + 1.6\pi, -\frac{3\pi}{2} + 2.4\pi\right) \tag{50}$$

$$t_3 \sim \mathcal{U}\left(-\frac{3\pi}{2} + 2.6\pi, -\frac{3\pi}{2} + 3.0\pi\right) \tag{51}$$

$$t \in \{t_0, t_1, t_2, t_3\} \tag{52}$$

$$h \sim \mathcal{U}(0, 2) \tag{53}$$

$$\mathbf{X} = \begin{bmatrix} \sin(t) \\ h \\ \text{sign}(t) \cdot (\cos(t) - 1) \end{bmatrix} + \epsilon \tag{54}$$

**Parallel Swiss Rolls**

Defining a raw Swiss Roll:

$$t \sim \mathcal{U}\left(\frac{3\pi}{2}, 3\pi\right) \tag{55}$$

$$h \sim \mathcal{U}(0, 30) \tag{56}$$

$$\mathbf{X}^{(\text{raw})} = \begin{bmatrix} t\cos(t) \\ h \\ t\sin(t) \end{bmatrix} + \epsilon, \tag{57}$$

$$\tag{58}$$

$$\tag{59}$$

the dataset consists of:

$$\mathbf{X}_2 = \mathbf{X}^{(\text{raw})} + \begin{bmatrix} 0 \\ 60 \\ 0 \end{bmatrix} \tag{60}$$

$$\mathbf{X} = \mathbf{X}^{(\text{raw})} \cup \mathbf{X}_2 \tag{61}$$

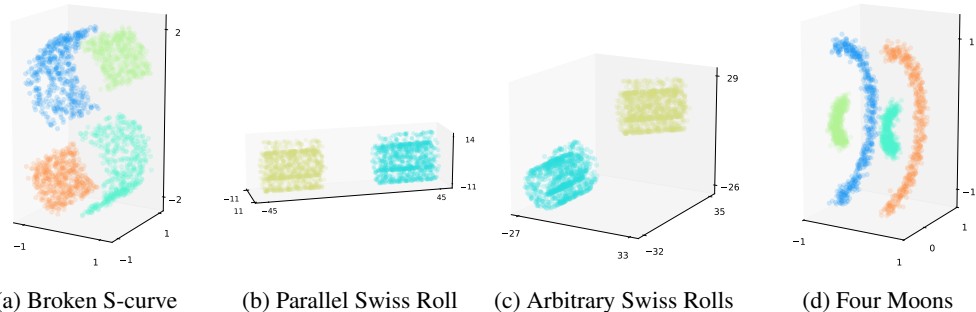

(a) Broken S-curve    (b) Parallel Swiss Roll    (c) Arbitrary Swiss Rolls    (d) Four Moons

Figure 3: Visual representation of the synthetic datasets used.

**Arbitrary Swiss Rolls**

Alongside the last, using the same original datasets:

$$t \sim \mathcal{U}\left(\frac{3\pi}{2}, 3\pi\right) \tag{62}$$

$$h \sim \mathcal{U}(0, 30) \tag{63}$$

$$\mathbf{X}^{(\text{raw})} = \begin{bmatrix} t\cos(t) \\ h \\ t\sin(t) \end{bmatrix} + \epsilon, \tag{64}$$

$$\tag{65}$$

$$\tag{66}$$

we apply the transformation:

$$\mathbf{R} = \begin{bmatrix} \cos(-\pi/4) & -\sin(-\pi/4) & 0 \\ \sin(-\pi/4) & \cos(-\pi/4) & 0 \\ 0 & 0 & 1 \end{bmatrix} \tag{67}$$

$$\mathbf{X}_1 = \mathbf{R}\mathbf{X}^{(\text{raw})} + \begin{bmatrix} 20 \\ 20 \\ 30 \end{bmatrix}, \tag{68}$$

and define the dataset as:

$$\mathbf{X}_2 = \mathbf{X}^{(\text{raw})} + \begin{bmatrix} 0 \\ -20 \\ 0 \end{bmatrix} \tag{69}$$

$$\mathbf{X} = \mathbf{X}_1 \cup \mathbf{X}_2 \tag{70}$$

**Four Moons**

Defining a raw pair of moons as:

$$t \sim \mathcal{U}(0, \pi) \tag{71}$$

$$\mathbf{X}_1 = \begin{bmatrix} 0 \\ \sin(t) \\ \cos(t) \end{bmatrix} + \epsilon \tag{72}$$

$$\mathbf{X}_2 = \begin{bmatrix} 0 \\ \frac{1-\sin(t)}{4} \\ \frac{-\cos(t)}{4} \end{bmatrix} + \epsilon \tag{73}$$

$$\mathbf{X}^{(\text{raw})} = \mathbf{X}_1 \cup \mathbf{X}_2, \tag{74}$$

the full dataset is defined:

$$\mathbf{X} = \mathbf{X}^{(\text{raw})} \cup \left(\mathbf{X}^{(\text{raw})} + \begin{bmatrix} 1 \\ 0 \\ 0 \end{bmatrix}\right) \tag{75}$$