# OpenReview forum: "Maximum Variance Unfolding on Disjoint Manifolds"
_ICLR.cc/2026/Conference — Submitted to ICLR 2026_

### Official Review · Reviewer_GwAJ · 2025-10-22

**Soundness:** 2
**Presentation:** 1
**Contribution:** 2
**Rating:** 2
**Confidence:** 5

**Summary:**

This paper presents an extended MVU algorithm to solve the graph-disconnectivity problem. While the problem is indeed interesting and the proposed approach seems reasonable, the paper misses important literature on building the neighborhood graph using algorithms like minimum spanning trees. And there are serious issues with the presentation: the paper seems disorganized and lacks careful proofreading. Please see my more detailed comments in the "weaknesses" section.

**Strengths:**

* The research problem is interesting
* The proposed method seems reasonable
* Multiple datasets are tested

**Weaknesses:**

* The presentation of this paper has serious issues. After reading the introduction, it's unclear to me what the technical contributions are. The authors only briefly mentioned them before the last paragraph; I suggest the authors highlight them and elaborate on them.

* The paper seems incomplete. At the beginning of Section 3, authors said "We provide a notation (glossary? variable index?) table in Appendix TODO."

* All the LaTex double-quotes are misused.

* Also the first few paragraphs of Section 3 are confusing: Why do we have six number items right after Algorithm 1?

* The chosen baselines are questionable. Except ENG being published in 2018, all others were published in 2006 or earlier. I also recall that autoencoder was one dimension-reduction method; why is it not even mentioned in the paper?

* Section 5 looks strange. Is it part of the evaluation?

* I also recall there were existing methods to solve the graph disconnectivity problem: instead of applying k-NN over the vertices, we could construct a minimum spanning tree (MST), or something similar that is connected, over the vertices. The authors didn't discuss this.

**Questions:**

* Please check the literature of applying minimum spanning trees and other data structures to construct connected neighborhood graphs.

* Please carefully proofread your manuscript.

---

### Official Review · Reviewer_zAf4 · 2025-10-27

**Soundness:** 1
**Presentation:** 2
**Contribution:** 2
**Rating:** 2
**Confidence:** 5

**Summary:**

The paper proposes Maximum Variance Unfolding on Disjoint Manifolds (MVU-DM), an extension of the classical Maximum Variance Unfolding (MVU) algorithm to settings where the knn graph is disconnected. The idea is to run MVU independently on each connected component, choose a small set of “representative points” per component, perform a global MVU only on those representatives (Eqs. 8–11), and then reposition the remaining points through an affine transformation estimated from their representatives (Step 6, Algorithm 1). The stated goals are to (i) enable MVU to handle disjoint manifolds, (ii) reduce computational cost, and (iii) preserve MVU’s local-isometry property.

**Strengths:**

The paper tries to address a genuine and practically relevant limitation of MVU, its inability to handle disconnected neighbourhood graphs, and proposes a conceptually simple strategy to overcome it. The method is easy to implement, with a clear algorithmic structure that could be integrated into existing MVU frameworks. The presentation is generally well organised and accessible, and the authors have made an effort to ensure experimental reproducibility through a comprehensive evaluation on both synthetic and natural datasets. The empirical results, while not sufficient to counterbalance the theoretical weaknesses, do provide suggestive evidence that the proposed decomposition can lead to improved efficiency in practice.

**Weaknesses:**

The data-generation assumptions are unstated and ambiguous.  The mathematical formulation is incomplete: centring and feasibility are not guaranteed. The affine alignment step undermines the core isometry guarantee. Representative and link selections are heuristic and theoretically unsupported. Computational advantages are empirically observed but not analytically justified. Details blow.

**Questions:**

There are some major theoretical concerns shown below.

1. What is the foundational assumptions? The paper alternates between describing the setting as “disjoint manifolds” and as “disconnected neighbourhood graphs”. It is therefore unclear whether each component is intended to correspond to a distinct manifold or whether disconnection merely results from a small k or sparse sampling. Moreover, there is no stated assumption about manifold separation, reach, or noise that would justify the construction of cross-component links or the definition of representative points.
Without explicit assumptions, the method’s applicability is ambiguous: it is not clear whether it models multiple manifolds, a single manifold with missing bridges, or simply artefacts of the neighbourhood graph.

2.  The convex MVU formulation (Eqs. 4–7 / Appendix B) uses the constraint \sum_{i,j}K_{ij}=0, which is weaker than the standard condition K\mathbf{1}=0 and \mathbf{1}^\top K=0. The double-sum constraint can hold even when K\mathbf{1}\neq0, meaning the embedding need not be centred and the variance-maximisation objective may become ill-posed. Appendix B repeats this error and introduces further typographical inconsistencies e.g. G vs K, mis-indexed terms. The convex relaxation no longer faithfully reproduces the classical MVU, and guarantees of equivalence and boundedness therefore do not hold.

3. The global MVU in Eqs. 8–11 constrains intra-component distances to equal those in the local MVU embeddings Y_p, while constraining inter-component distances to equal those in the original space X. These two distance metrics are only related up to unknown rigid motions and possible numerical distortions. The paper provides no theoretical argument that a single Euclidean configuration Z exists that satisfies both constraint sets simultaneously. In general, such hybrid distance systems are inconsistent unless the underlying constraint graph is globally rigid. A theoretical proof of feasibility or rigidity is desirable, or alternatively a relaxation with bounded-error constraints should be proposed.

4. Eq (8)–(11) describe only the optimisation on representative points; the affine transformation used to map the remaining points appears solely in Step 6 as a narrative statement “representing points in homogeneous coordinates and computing an affine transformation matrix”. No explicit formulation or orthogonality restriction is given. Because a general affine transformation can introduce scale and shear, it violates MVU’s local-isometry guarantee for the rest of the data. The claim that there is “no loss of information” (p. 4) is therefore unsubstantiated. The authors should either constrain this step to a rigid Procrustes alignment or prove analytically that the affine maps are near-isometric.

5. Representatives are defined as extrema along principal directions of the local embeddings |Z_p|=2d_p. There is no result proving that such points determine each component’s rigid motion uniquely or that the procedure is numerically stable. For curved or branched manifolds, principal-direction extrema can be unstable under noise and can lead to degenerate configurations. A conditioning or rigidity analysis would be required to justify this step, ideally linking the selection to an optimality or stability criterion.

6. Beware that inter-component link construction can introduce shortcuts. Step 4 connects components by their nearest cross-pair in the original space until the graph is connected. This procedure can create non-geodesic shortcuts between manifolds that are spatially close but genuinely separate, distorting the geometry MVU is supposed to preserve. No theoretical justification or safeguard is offered against this. A geometric separation condition or an adaptive link penalty should be introduced to avoid such artefacts.

7. Although Eq (8)–(11) superficially resemble the vanilla MVU formulation, they operate only on a small set of representative points |Z|=\sum_p|Z_p|, not on the full dataset of N samples. Each component MVU of size |X_p| can be solved in parallel, and the global SDP is defined on the much smaller matrix Z, explaining the empirical speed-ups reported in Table 4. However, the paper provides no formal complexity analysis or explicit count of constraints. A quantitative comparison of computational cost and asymptotic behaviour should be added.

8. In Appendix B the rank constraint is dropped to achieve convexity, but no discussion is given of when this relaxation remains tight under the new mixed-distance constraints. Even in the standard connected case, the tightness of the MVU relaxation is subtle; without an analysis of the mixed system, it is unclear whether the obtained Gram matrix truly corresponds to a low-dimensional embedding.

Minor issues

Notation is inconsistent in several places, e.g. G vs K, x_l vs x_j, which undermines confidence in the derivations. The stated complexity O((kN)^3) (p. 2) conflates graph sparsity with the semidefinite cone dimension and should be clarified. The paper also lacks qualitative visualisations to illustrate the reconstructed geometry and verify that manifold relationships are preserved. The review of some methods such as HLLE in Appendix A seems not used anywhere. There are unfinished tasks (TODO) in the paper. Should tidy it up before submission.

---

### Official Review · Reviewer_ij7F · 2025-10-30

**Soundness:** 2
**Presentation:** 3
**Contribution:** 2
**Rating:** 2
**Confidence:** 5

**Summary:**

This paper extends MVU, a graph-based method for nonlinear dimensionality reduction that retains local isometry, to
 the common case where data lie on disjoint manifolds. MVU is applied to each data component separately, and a set of representative points is chosen for each component, and finally a neighborhood graph between components is constructed using these representative points and MVU is performed on this neighborhood graph. Experiments are conducted on some toy and simple datasets.

**Strengths:**

1. Easy to follow.
2. Computational time and memory requirement are decreased as compared with some baselines.

**Weaknesses:**

1. Novelty and contributions are limited. The proposed method is basically a simple extension of MVU.
2. The robustness of the proposed algorithm is not discussed.
3. The compared baselines are all proposed before the deep learning era, and the datasets used in experiments are also somewhat outdated.
4. line 161, notation(glossary? variable index?) table in Appendix TODO, which is absent in this paper.
5. In ACKNOWLEDGMENTS, the sentence " This work was partly funded by the Foundation of Science and Technology through scholarship
 2024.04726.BD." should be removed.
6. The appendices A and B are reviews of some existing works.

**Questions:**

The notation "$\textbf{y}_{p,i}$" represents the embedding of representative points. This should be stated explicitly in the presentation.

---

### Official Review · Reviewer_yCsb · 2025-11-04

**Soundness:** 2
**Presentation:** 3
**Contribution:** 3
**Rating:** 2
**Confidence:** 4

**Summary:**

This paper proposes Maximum Variance Unfolding on Disjoint Manifolds (MVU-DM), an extension of the classic nonlinear dimensionality reduction method Maximum Variance Unfolding (MVU), specifically designed to handle data that lies on disconnected manifolds.

**Strengths:**

1. It can effectively embed data from multiple, separate manifolds.
2. By breaking the problem into smaller, parallelizable sub-problems and using a representative subset for the final global step, it reduces computation time and memory usage.
3. It maintains MVU's desirable property of preserving local distances (strong local isometry).
4. The method does not introduce any additional hyperparameters beyond those of standard MVU.

**Weaknesses:**

1. The optimization problem becomes unbounded, as disconnected components can be moved infinitely far apart to maximize variance.
2. High computational cost for large datasets, as MVU involves solving a large Semidefinite Program (SDP).
3. The work lacks practical significance.
4. The work is not suitable for big data processing.

**Questions:**

1. The optimization problem becomes unbounded, as disconnected components can be moved infinitely far apart to maximize variance.
2. High computational cost for large datasets, as MVU involves solving a large Semidefinite Program (SDP).
3. The work lacks practical significance, compared with deep learning methods or other works.
4. Also, the work is not suitable for big data processing.

---

### Meta-Review · Area_Chair_nzFV · 2026-01-06

**Summary:**

This paper extends the maximum variance unfolding method to cases where the underlying graph is disconnected. Reviewers critized the paper for its writing, novelty and restricted experiments, and they unanimously recommend rejection. The authors did not provide a rebuttal, and I thus also recommend rejection.

**Reviewer Concerns:**

All the concerns mentioned in the summary are unaddressed.

**Reviewer Scores:**

No reviewer would have changed their score given there was no rebuttal.

---

### Decision · Program_Chairs · 2026-01-26

Reject